# Tempo-spatial variations of zooplankton communities in relation to environmental factors and the ecological implications: A case study in the hinterland of the Three Gorges Reservoir area, China

Bo Lan[1,2], Liping He[2], Yujing Huang[3], Xianhua Guo[2], Wenfeng Xu[2], Chi Zhu[4,5]*

1 Research Center for Sustainable Development of the Three Gorges Reservoir Area, Chongqing Three Gorges University, Chongqing, China, 2 College of Environment and Chemical Engineering, Chongqing Three Gorges University, Chongqing, China, 3 College of Chinese Traditional Medicine, Chongqing Three Gorges Medical College, Chongqing, China, 4 Jiangsu Environmental Engineering Technology Co. LTD, Nanjing, China, 5 Jiangsu Provincial Academy of Environmental Science, Nanjing, China

* zhuchi126@126.com

**Data Availability Statement:** All relevant data are within the manuscript and its Supporting information files.

## Abstract

To expand the knowledge on the tempo-spatial patterns of zooplankton and the key modulated factors in urban aquatic ecosystem, we investigated zooplankton and water quality from April 2018 to January 2019 in the hinterland of the Three Gorges Reservoir area, Wanzhou City of China. The results indicated that water quality indicated by the trophic state index (TSI) reached a state of mesotrophication to light eutrophication in the Yangtze River, and a state of moderate- to hyper- eutrophication in its tributaries. Based on the biomass of zooplanktons, *Asplanchna priodonta* was the most common specie in April; *Encentrum* sp., *Filinia cornuta* and *Epiphanes senta* were the most noticeable species in summer; Cyclopoida Copepodid, *Sinocalanus dorrii* and *Philodina erythrophthalma* became the dominant species in winter. Generally, rotifers prevailed in April and August, and copepods became the most popular in January. According to canonical correspondence analysis, nitrate, temperature (T), ammonia, water level and permanganate index ($COD_{Mn}$) significantly influenced the community structure of zooplankton ($p < 0.05$). The dominant species shifts of zooplankton were partly associated with nutrient level (nitrate and ammonia) under periodic water level fluctuations. Rotifers and protozoans were characterized as high T adapted and $COD_{Mn}$-tolerant species comparing with cladocerans and copepods. The ratio of microzooplankton to mesozooplankton ($P_{micro/meso}$) has presented a strongly positive relationship with T ($p < 0.001$), as well as $P_{micro/meso}$ and $COD_{Mn}$ ($p < 0.001$). It implied that zooplankton tended to miniaturize individual size via species shift under high T and/or $COD_{Mn}$ conditions induced by global warming and human activities. The information hints us that climate change and human activities are likely to produce fundamental changes in urban aquatic ecosystem by reorganizing biomass structure of the food web in future.

**Funding:** This work was supported by the National Natural Science Foundation of China (41902024); the Program of Chongqing Science and Technology Commission (cstc2019jcyj-msxmX0656); the Science and Technology Research Program of Chongqing Municipal Education Commission (KJQN201801221); Talent Introduction Program of Chongqing Three Gorges University (17RC08); the Research Center for Sustainable Development of the Three Gorges Reservoir Area (18sxxyjd12). All grants were awarded to B.L. The funders had no role in study design, data collection and analysis, decision to publish, or preparation of the manuscript.

**Competing interests:** C.Z. has financial interests in Jiangsu Environmental Engineering Technology Co. LTD. The remaining authors have declared that no competing interests exist. This does not alter our adherence to PLoS ONE policies on sharing data and materials.

## Introduction

Aquatic ecosystem serves plenty of ecosystem functions for human beings, however, it has witnessed varying degrees of water environment changes (e.g., flow regime changes, contamination and so on) induced by anthropogenic activities and climate changes during the past decades [1–3]. In particular, urban aquatic ecosystems were more likely to be disturbed by intensive anthropogenic activities directly and/or indirectly, for instance, the water body along the Yangtze River Economic Zones of China has documented to be disturbed by the single or combined effects of multiple environmental incentives [4, 5]. Specifically, due to the construction of the world's largest regulating reservoir, named the Three Gorges Reservoir (TGR) situated on Yangtze River, the water level (WL) has presented periodic fluctuations within a calendar year. The variations of hydrological regime, superimposed on pollutants input originated from urban sewage, industrial wastewater, agricultural fertilizers, leaching of soil organic matters, etc. [6, 7], resulted in significant changes in aquatic environment, such as TGR impounding resulted prolonged retention time, materials transport and the associated biochemical processes in urban aquatic ecosystems of the TGR [4, 8]. Thus, the relating environment problems, such as soil erosion, heavy metals accumulation, large-scale habitat alterations, degraded water quality, algae bloom, variations of biotic community structure and so on, have been substantially reported [4, 7, 9, 10]. In conclusion, the urban aquatic environment in the Three Gorges Reservoir area (TGRA) has undergone significant changes, profoundly affecting biotic components (e.g., zooplankton) living in urban aquatic ecosystems ultimately.

In order to better protect urban aquatic ecosystem, variations of aquatic biotic components affected by diverse environmental stressors play vital roles in fully understanding the regulatory mechanisms within aquatic ecosystem disturbed by climate changes and anthropogenic activities. Zooplankton as an important component of aquatic ecosystem, not only feeds on phytoplankton, but also provides essential food resources for fishes, thus it fulfills important ecological functions of aquatic ecology [10–12]. Zooplankton is featured of rich species, typically small in size with short reproductive times, high-sensitivity to local water environment, making themselves vital study objects in aquatic ecology. It usually consists of protozoans, rotifers, cladocerans and copepods. At present, lots of research regarding zooplankton in theTGRA mainly focus on the community characteristics of zooplanktons since the impounding of the TGR, such as the community structure of copepods and cladocerans in the Yangtze River [13] and the seasonal succession characteristics of zooplankton in Pengxi River [14], spatial-temporal distribution and vertical migration of zooplankton in Xiangxi River within the TGRA [11, 15]. However, the seasonal succession in the urban aquatic ecosystem and the relationship between zooplankton communities and environmental factors has not been substantially studied.

The paper selected Wanzhou City which is situated in hinterland of the TGRA as a typical study area, aimed to (i) understand present situation of water pollution status in urban aquatic ecosystem of Wanzhou City; (ii) reveal the spatio-temporal distribution patterns of zooplankton after the TGR impoundment; (iii) investigate the associated ecological succession processes and the key environmental factors shaping the community structures; (iv) analyze the implications of those environmental factors effecting the zooplankton in future. Those results will deepen our comprehension on zooplankton community biology in riverine ecosystem under the effects of climate changes and anthropogenic disturbances.

## Methods

### Study area

Wanzhou City is located within the hinterland of the Three Gorges Reservoir area (TGRA) of Chongqing, China (Fig 1A). As the second largest city of Chongqing, it is known as an

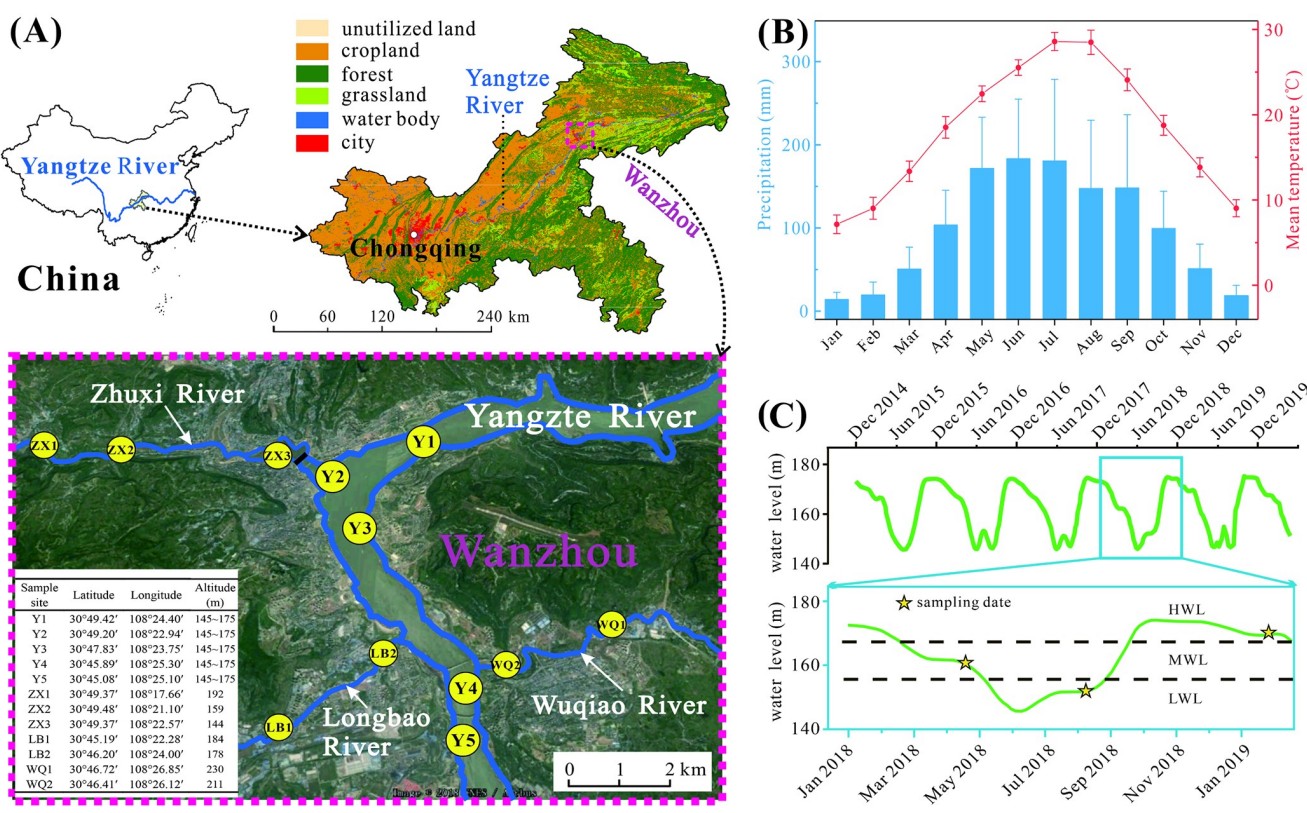

**Fig 1. Sampling sites and the hydro-climatical information.** (A) Sampling sites were located in Wanzhou City of Chongqing, as well as the land utilization of Chongqing in 2018 for background reference; (B) The monthly mean temperature and precipitation of Wanzhou City; (C) The water level fluctuations in recent five years with zoom-in details from January 2018 to January 2019.

immigrant city where a large number of residents from surrounding areas were relocated in due to the construction of the TGR. In 2020, the urban population in Wanzhou City is already more than one million as massive people immigrated to the city (http://www.wz.gov.cn). Wanzhou City is also a typically riverine city along the Yangtze River which is known as the largest river in China. Quite a few secondary tributaries of the Yangtze River come through Wanzhou City, including Zhuxi River (ZX), Longbao River (LB) and Wuqiao River (WQ). Thus, those river networks constitute a pollutant repository and drainage pathway because a large quantity of industrial and domestic sewage discharged from Wanzhou City. Considerable environmental pressures to the local aquatic ecosystem can be predictable, potentially threatening the regional ecological security of water body within the TGRA.

The local climate prevails subtropical monsoon climate, with a mean annual temperature of 18.3 ± 0.6 ˚C and an average annual precipitation of 1187.8 ± 208.7 mm based on the instrumental-recorded data during 1954–2019 (data source: http://data.cma.cn/). The highest mean monthly temperature is 28.6 ± 1.1 ˚C in July and the lowest is 7.2 ± 1.1 ˚C in January. The precipitation mainly focuses on warm season, accounting for ~70% of the annual precipitation (Fig 1B). Therefore, the high precipitation in warm season frequently gives rise to flood events, resulting in turbid water with high suspended sediment loads in Yangtze River [4].

The hydrological ecosystem in Wanzhou City is profoundly affected by the TGR nowadays. In details, an annually cyclic water-level modulating strategies is adopted to conduct the TGR operation, impounding clear water in dry season (i.e., approximately from October to

February of the following year) and discharging turbid water in wet season (i.e., approximately from May to September). Due to the operating strategies of TGR, the WL of Yangtze River with TGRA is characterized by annually cyclic water-level variations superimposed by additional changes caused by rainfall events during the wet season [4]. During last decade, it has witnessed a middle water-level (MWL) phase during March to May with slowly descending trend, a low water-level (LWL) phase during June to August with the lowest water-level of 145 m around June, a MWL phase in September with rapidly climbing trend and a high water-level (HWL) phase during October to February of subsequent year with the highest water-level of 175 m around December (Fig 1C). The water-level variation imposes different effects on the tributaries of Yangtze River, and creates an extensive water level fluctuation zone (WLFZ) of the TGR, being subject to intensive anthropogenic disturbances and natural impacts (e.g., precipitation variations), and thus resulted in prevalence of annual plants and periodic biochemical cycles in WLFZ [4, 7].

## Sampling procedures and data measurement

A total of 36 pairs water samples and zooplankton samples were collected at 12 sites (5 sites in the Yangtze River and 7 sites in its tributaries) in Wanzhou City in three seasons (Fig 1C), i.e., April 2018 (indicating spring at MWL), August 2018 (referring to summer at LWL), January 2019 (corresponding to winter at HWL). The location information of sample sites can refer to Fig 1 after taking account of the spatial distribution of river networks across the urban city of Wanzhou. Samples were conducted at the backwater zone of each river. Water samples were collected at depths of 20 cm by using polymethyl methacrylate hydrophore and stored in an incubator with ice packs for physicochemical analyses of water quality in library. Zooplankton samples were collected from mixed water sample derived from top, middle and bottom layers of water column, depending on the water depth when sampling. 15–30 L vertical mixing water sample was filtered using 64 μm nylon mesh net and concentrated to 30 ml for quantitative analysis of cladocerans and copepods exclusively. And 1.5 L mixed water sample was placed in separatory funnel motionlessly for two days after fixed by Lugol's iodine solution, discarded the supernatant to obtain 30 ml concentrate for identifying protozoans and rotifers quantitatively. Formaldehyde solution was added to each zooplankton sample for long-term preservation. Zooplankton identification was referred to relevant literatures [16–19].

Water temperature (T), pH and dissolved oxygen (DO) were instrumentally measured *in situ*. Other water quality parameters, including total nitrogen (TN), dissolved total nitrogen (DTN), nitrate ($NO_{3^-}$-N), nitrite ($NO_{2^-}$-N), ammonia ($NO_{4^+}$-N), total phosphorus (TP), dissolved total phosphorus (DTP), active phosphorus (SRP), permanganate index ($COD_{Mn}$) and total suspended solids (TSS) were referred to standard methods drafted by American Public Health Association [20]. The chlorophyll a (Chl a) was extracted by 90% acetone and measured by spectrophotometer after filtration of a 500–1000 mL water sample through GF/C filters, according to the standard methods [20]. For the quality assurance and quality control programs of nutrient determinations, two solvent blanks, two procedure blanks, two matrix spikes, sample duplicate and recovery tests were performed for each batch of twelve water samples. The averaged recovery of standard addition varied between 95% ~105% and the relative standard deviation was better than 5%. Each calibration curve for the nutrient quantification was required a good linearity ($R^2 > 0.999$). The nutrient content of natural water body is normally higher than lower limit of detection, possibly higher than upper limit of detection in each assay method, thus dilution to the proper concentration before measurement if needed. $COD_{Mn}$ is an indicator of organic pollutants and was determined using the titration method after oxidation treatment by potassium permanganate.

## Data analysis

The trophic state index (TSI) was employed to describe the trophic state of water quality. The relating formulas can refer to Jin and Tu [21]. According to the standard trophic categories, $30 \leq TSI < 50$ indicated mesotrophic; $50 \leq TSI < 60$ denoted for light eutrophication; $60 \leq TSI < 70$ represented moderate eutrophication; $70 \leq TSI$ was regarded as hyper-eutrophication [22].

The tempo-spatial analysis of zooplankton summarized all samples in each river for comprehensive comparisons of seasonal changes by using biomass data based on the consideration that biomass data may be more important in fishery resources with respect to density. The bio-index, including the biodiversity index (Shannon-Weiner index, $H'$), species richness indexes (Simpson index, $D$) and evenness index (Pielou index, $J$), were employed to assess the status of community structure. The calculation formulas were listed as follows [10]:

$$H' = -\sum P_i \ln P_i$$

$$D = (S - 1)/\ln N$$

$$J = H'/\ln S$$

where $P_i$ is the ratio of the number of the $i$th species to the total number ($N$) in each sample, $S$ is the number of species in each sample.

Generally, protozoans and rotifers are usually categorized to microzooplankton, whereas cladocerans and copepods are usually classified to mesozooplankton. Thus, the index $P_{micro/meso}$, i.e., the proportion between microzooplankton and mesozooplankton based on density or biomass data, was introduced to reflect the size change of zooplankton, i.e.,

$$P_{micro/meso} \text{ based on density} = density_{protozoans+rotifers}/density_{cladocerans+copepods}$$

$$P_{micro/meso} \text{ based on biomass} = biomass_{protozoans+rotifers}/biomass_{cladocerans+copepods}$$

A detrended correspondence analysis (DCA) was performed to extracted gradient by using biomass dataset of zooplankton species in this survey firstly. Because DCA gradient was $>3$, multivariate statistical analysis was adopted the recommended unimodal model, i.e., canonical correspondence analysis (CCA). CCA can intuitively elucidate the relationships among zooplankton compositions, samples and environmental factors within a two-dimensional coordinate system, detecting the most important physicochemical variables shaping the biotic community [23]. In CCA procedure, the biomass of species and the environmental factors were root transformed in order to reduce the variance differences of extremums and remove dimensionality. Subsequently, the environmental factors with inflation factor greater than 10 were excluded from CCA procedure in order to reduce the multicollinearity among environmental factors. Then, interactive-forward-selection was used to further screen the environmental variables which were significantly related to the ordination axis by using Monte Carlo permutations ($p < 0.05$, 999 iterations). CCA ordination was performed using Canoco 5.0 [24]. Spearman rank correlation analysis was used for correlation analysis, and one-way ANOVA with Dunnett's C tests were employed to test the significances among the environmental variables and biotic data (e.g., density and biomass). The significance level was considered as significant and extremely significant according to $p < 0.05$ and $p < 0.01$, respectively. Data analysis was done in SPSS 22.0.

## Results

### Water environment status

Physicochemical data of water quality showed that nearly all indexes, including T, DO, $NO_3^-$-N, SRP, $COD_{Mn}$, Chl a and TSS, were of extremely significant tempo-spatial differences except TN and DTN ($p < 0.001$, Table 1). Generally, the nitrogen and phosphorus concentrations in the Yangtze River were lower than its tributaries, indicating that the former registered a better water quality status than the latter. In addition, $COD_{Mn}$ ranged from 1.6 to 14.6 mg/L with the lowest value of 1.9 ± 0.5 mg/L in the Yangtze River in April and the highest value of 12.6 ± 2.0 mg/L in ZX River in August. Chl a of ZX River recorded a significantly higher level than the others correspondingly, whereas TSS of the Yangtze River in August at LWL was significantly higher than that of other rivers at the same period ($F = 5.27$, $p = 0.027$).

The water quality evaluation based on TSI has exhibited obvious tempo-spatial characteristics (Fig 2). For the Yangtze River, TSI in August at LWL (60.6±5.1) belonged to the middle eutrophication, while TSI in April at MWL and in January at HWL (46.3±3.3 and 46.9±1.8, respectively) both belonged to mesotrophic status. Statistically, TSI in August was significantly higher than the other two ($F = 24.80$, $p < 0.001$). For the tributaries, TSI was categorized as middle to hyper eutrophication, ranged from 60.2±5.6 to 71.4±5.1, and the average TSI was relatively higher in August than the other two seasons, similar with the Yangtze River. However, there were no significant differences among TSI values of tributaries at different sampling periods statistically ($F = 1.60$, $p = 0.22$). In general, the water quality of the Yangtze River was better than its tributaries in Wanzhou City of the TGRA.

**Table 1. Physical and chemical parameters of water quality in the Yangtze River and its tributaries.**

| Environmental parameters | April 2018 (MWL) | | | | August 2018 (LWL) | | | | January 2019 (HWL) | | | | p |
|---|---|---|---|---|---|---|---|---|---|---|---|---|---|
| | Y | ZX | LB | WQ | Y | ZX | LB | WQ | Y | ZX | LB | WQ | |
| T (°C) | 18.4±0.5 | 23.7±1.4 | 23.3 ±1.8 | 19.5 ±1.2 | 25.9±1.3 | 32.8±0.6 | 26.8 | 28.2±0.9 | 10.1±0.04 | 10.0±0.1 | 10.3 ±0.07 | 10.5 ±0.07 | **0.000** |
| pH | 8.2±0.1 | 8.3±0.3 | 8.3±0.3 | 8.1±0.2 | 7.9±0.1 | 8.6±0.5 | 8.0±0.3 | 7.9±0 | 8.2±0.1 | 8.2±0.3 | 7.9±0.3 | 7.9±0.1 | **0.035** |
| DO (mg/L) | 9.0±0.6 | 10.7±3.4 | 7.5±0.2 | 7.8±1.3 | 8.4±0.9 | 12±1.3 | 7.7±2.8 | 6.9±0.5 | 12.1±0.3 | 7.8±1.0 | 8.5±0.4 | 10.0±0.3 | **0.000** |
| TN (mg/L) | 2.1±0.4 | 4.3±1.9 | 5.8±4.5 | 6.8±5.3 | 2.9±0.5 | 2.6±0.9 | 7.0±6.3 | 2.7±1.4 | 2.3±0.2 | 5.0±0.2 | 11.6 ±11.5 | 6.1±3.0 | 0.089 |
| DTN (mg/L) | 1.5±0.1 | 3.4±1.5 | 5.5±4.9 | 6.0±5.1 | 2.2±0.2 | 2.0±0.3 | 6.3±5.9 | 2.5±1.1 | 1.9±0.4 | 4.5±0.5 | 8.7±7.4a | 5.5±3.4 | 0.065 |
| $NO_4^+$ −N(mg/L) | 1.0±0.4 | 3.0±0.9 | 1.0±0.5 | 1.4±1.2 | 0.2±0.05 | 0.2±0.2 | 2.1±2.0 | 0.6±0.6 | 0.5±0.2 | 1.6±0.4 | 3.8±4.6 | 2.1±1.7 | **0.025** |
| $NO_3^-$ −N(mg/L) | 0.4±0.03 | 0.2±0.1 | 1.0±0.6 | 0.3 ±0.04 | 1.9±0.1 | 1.0±0.4 | 3.0±2.5 | 0.9±0.1 | 1.0±0.03 | 1.0±0.1 | 1.2±0.02 | 1.1±0.1 | **0.000** |
| $NO_2^-$ −N(mg/L) | 0.1±0.01 | 0.6±0.5 | 0.3±0.4 | 0.2±0.1 | 0.01 ±0.01 | 0.2±0.05 | 0.6±0.7 | 0.5±0.5 | 0.04 ±0.004 | 0.04±0.0 | 0.04±0.0 | 0.04±0.0 | **0.031** |
| TP (mg/L) | 0.2±0.03 | 0.4±0.1 | 0.5±0.1 | 0.7±0.5 | 0.5±0.1 | 0.4±0.1 | 0.5±0.2 | 0.5±0.4 | 0.13±0.01 | 0.4±0.1 | 0.7±0.3 | 0.5±0.3 | **0.013** |
| DTP (mg/L) | 0.1±0.01 | 0.3±0.01 | 0.3±0.1 | 0.7±0.6 | 0.2±0.1 | 0.2±0.1 | 0.4±0.2 | 0.4±0.3 | 0.1±0.01 | 0.3±0.1 | 0.6±0.4 | 0.4±0.2 | **0.011** |
| SRP (mg/L) | 0.05 ±0.01 | 0.2±0.03 | 0.2±0.1 | 0.4±0.5 | 0.1±0.01 | 0.1±0.1 | 0.4±0.3 | 0.4±0.3 | 0.1±0.004 | 0.3±0.1 | 0.5±0.3 | 0.3±0.2 | **0.005** |
| $COD_{Mn}$ (mg/L) | 1.9±0.5 | 8.0±3.5 | 5.0±1.3 | 6.9±3 | 9.0±1.2 | 12.6±2.0 | 9.8±2.5 | 11.6±1.9 | 3.6±0.8 | 5.1±0.5 | 9.7±3.8 | 7.2±0.8 | **0.000** |
| Chl a (μg/L) | 3.5±1.6 | 50±48.9 | 4.9±0.3 | 4.2±0.1 | 6.6±10.6 | 73.0 ±39.5 | 20.0 ±2.0 | 31.6 ±11.2 | 1.7±0.4 | 17.4 ±10.7 | 6.1±1.4 | 7.2±3.3 | **0.002** |
| TSS (mg/L) | 3.8±2.5 | 21.3 ±23.6 | 25.2 ±4.6 | 10.2 ±2.9 | 113±61.2 | 6.9±3.6 | 28.6 ±0.8 | 7.7±0.8 | 4.1±1.6 | 1.6±1.5 | 18.6±5.7 | 11.4 ±10.6 | **0.000** |

Note: bold p value indicated $p < 0.05$.

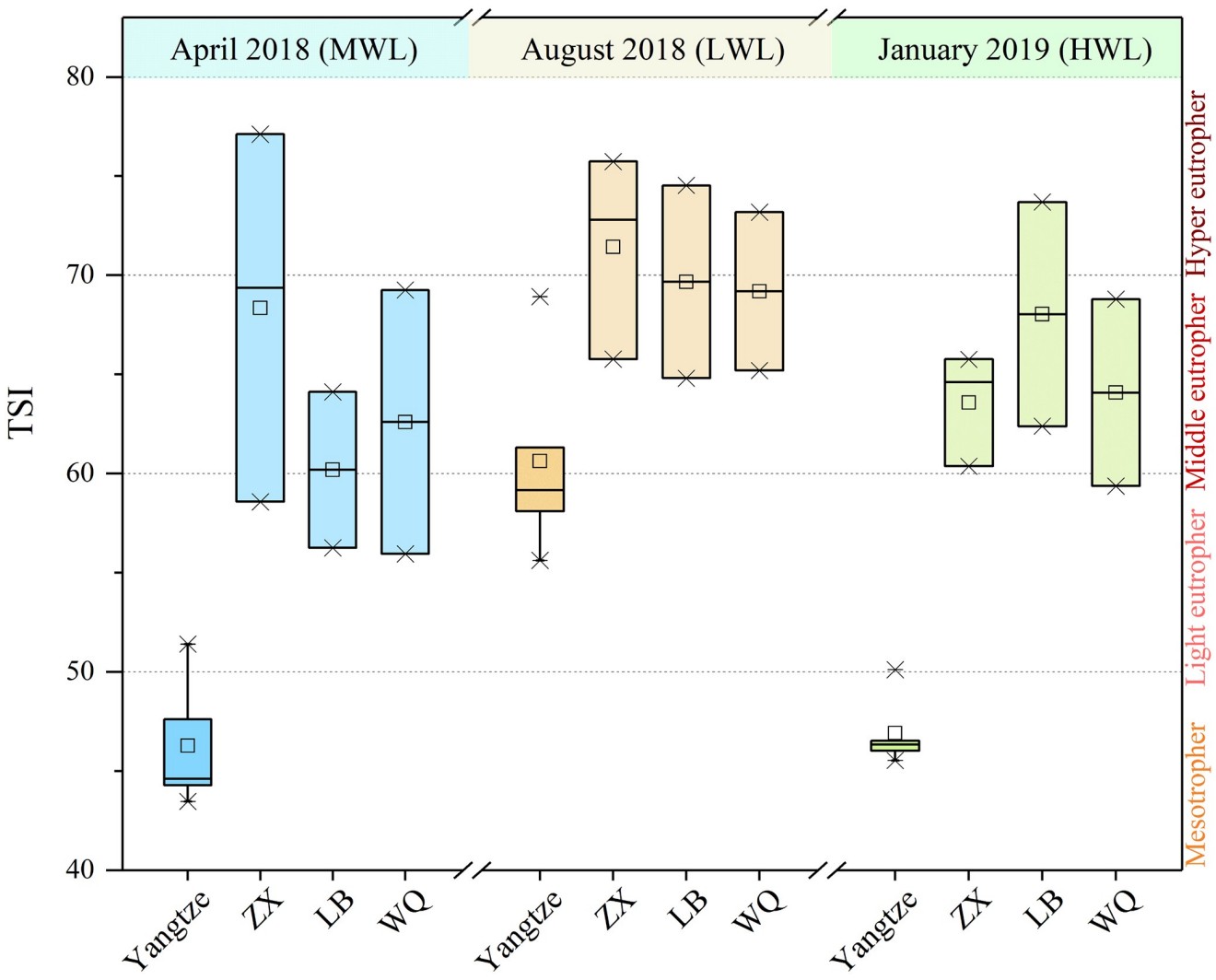

**Fig 2. TSI variations of the Yangtze River and its tributaries in Wanzhou City.**

## Zooplankton community structure

**Zooplankton composition and the temporal-spatial characters.** A total of 108 species and subspecies of zooplankton in 69 genera were identified in this study, belonging to protozoans, rotifers, cladocerans and copepods (S1 Table). There were several frequently occurring species in those samples, such as Ciliate, *Vorticella* sp., *Euchlanis dilatate*, *Synchaeta* sp., *Encentrum* sp., *Sinocalanus dorrii*, Canaloida Copepodid, Cyclopoida Copepodid and Nauplius (Fig 3 and S1 Table). Based on the heatmap of the zooplankton, the distribution of species density and biomass were visually displayed among all samples. For instance, *Vorticella* sp. (species NO. 28) was more abundant in the Yangtze River than its tributaries. Some species represented mismatch characters between density and biomass. The most obvious representatives were zooplankton larvae, such as Canaloida Copepodid, Cyclopoida Copepodid and Nauplius (species NO. 106–108), i.e., those components were not obvious in quantitative terms, however, they registered relative high biomass in August (Fig 3).

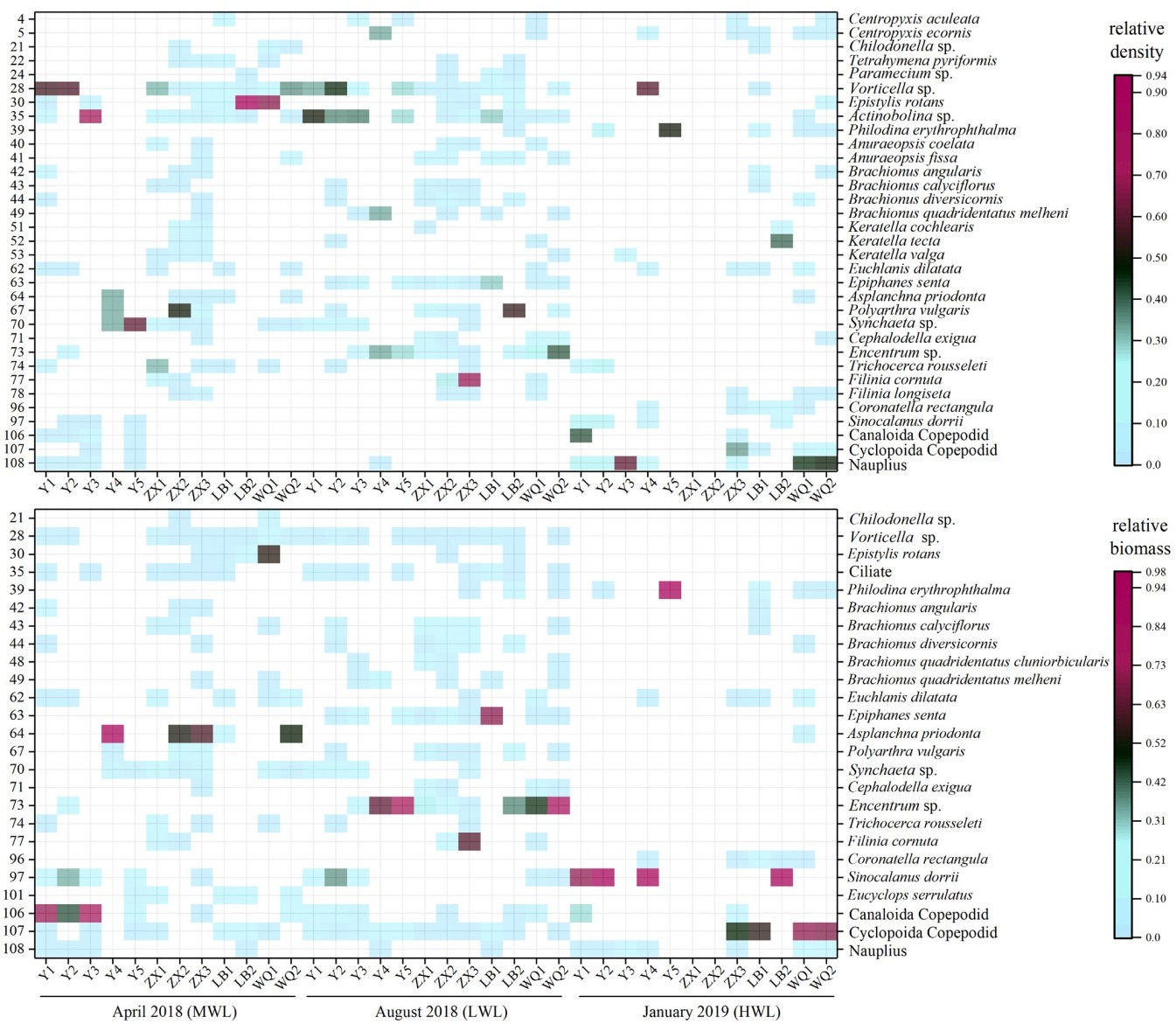

**Fig 3. Heatmap of zooplanktons species among the all samples.** The data was based on the relative density (above) and the relative biomass (below). Only showed those species with maximum proportion >5% and frequency ≥5 in all samples. The number on the Y axis was corresponding to that in S1 Table.

Substantial tempo-spatial variations of zooplankton were detected among the Yangtze River and its tributaries in Wanzhou City (Fig 4). Specifically, in April, Canaloida Copepodid (43.0%), *Asplanchna priodonta* (18.9%) and *S. dorrii* (16.7%) accounted for the top three in term of biomass proportion in the Yangtze River, respectively (note: the number in bracket denoted for the biomass proportion in each river, the same below). *A. priodonta* (37.7%) was the absolutely dominant specie in ZX River, *Eucyclops serrulatus* (22.6%), *A. priodonta* (12.1%) and Cyclopoida Copepodid (9.4%) was the dominant species of LB River, whereas *Epistylis rotans* (27.6%) and *A. priodonta* (25.3%) accounted for the top two percentages in WQ River. Generally, it was dominated by rotifers and copepods in water body. In summer, *Encentrum* sp. as a dominant specie, accounting for 31.8% in the Yangtze River and 63.7% in WQ River, respectively. *Filinia cornuta* (25.1%) and *Epiphanes senta* (34.5%) were the most noticeable

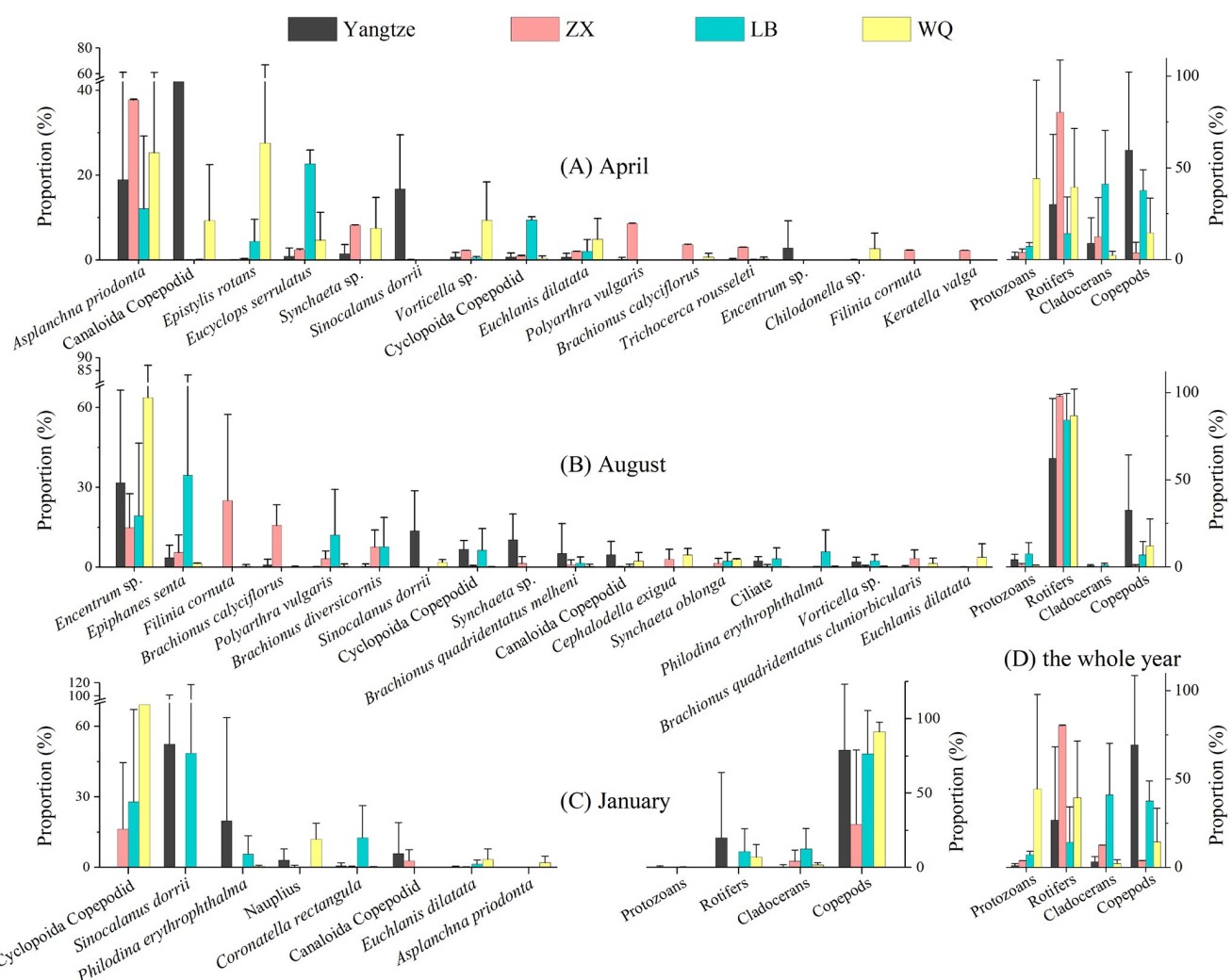

**Fig 4. Spatio-temporal patterns in zooplankton communities based on the biomass data.** The sampling season was April at MWL (A), August at LWL (B) and January at HWL (C), respectively. The four categories of zooplankton in the three seasons were also summarized (D). Only showed those species with average proportion >2% in each river and frequency ≥5. Yangtze = the Yangtze River, ZX = Zhuxi River, LB = Longbao River, WQ = Wuqiao River, The same below.

species in ZX River and LB River, respectively. It should be pointed out that rotifers were the absolutely dominant category in this season. In winter, Cyclopoida Copepodid, *S. dorrii* and *Philodina erythrophthalma* became the dominant species in the Yangtze River and its tributaries and copepods became the most popular category. Based on zooplankton categories in the whole year, the copepods (69.5%) and rotifers (26.6%) accounted for the highest two proportions in the Yangtze River. Rotifers (80.3%) were the absolutely dominant category in ZX River, cladocerans (41.0%) and copepods (37.7%) were the most important categories in LB River, and protozoans (44.3%) and rotifers (39.3%) were the dominant category in WQ River.

Significant seasonal shifts among dominant species have been observed in the study. In details, from April to August, *A. priodonta*, Canaloida Copepodid, *Simocephalus vetulus*, *E. rotans*, *E. serrulatus* declined in the numbers and shifted to the dominance of *Brachionus quadridentatus melheni*, *Brachionus calyciflorus*, *Schmackeria forbesi*, *Brachionus diversicornis*, *F. cornuta*, *E. senta* and *Encentrum* sp. (Fig 5). From August to January, those abundantly

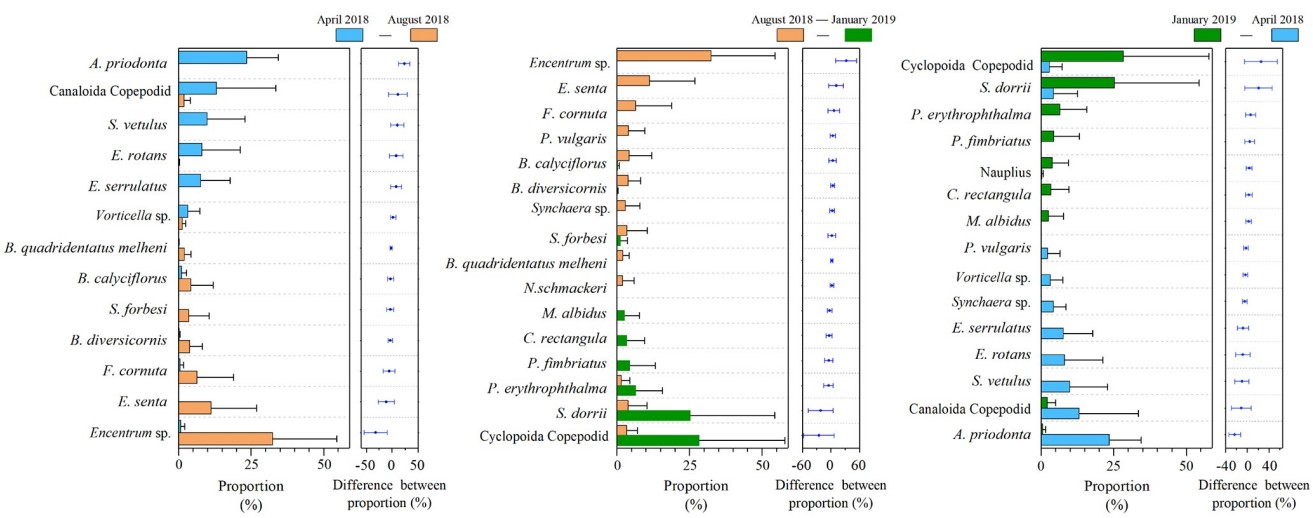

**Fig 5. The species shift of zooplankton in the three seasons.** Only showed the species those proportion differences were more than 2%.

appeared in August, such as *Encentrum* sp., *E. senta*, *F. cornuta*, *Polyarthra vulgaris*, *B. calyci-florus* shifted to the dominance of *Paracyclops fimbriatus*, *P. erythrophthalma*, *S. dorrii* and Cyclopoida Copepodid in January. From January to April, Cyclopoida Copepodid, *S. dorrii* and *P. erythrophthalma* obviously shifted to another species, such as *E. serrulatus*, *E. Rotans*, *S. vetulus*, Canaloida Copepodid and *A. priodonta*. It should be pointed out that the zooplankton assemblage in April of the subsequent year was assumed to be similar to that in April of this year under the effects of periodic WL fluctuation and climate condition, and thus actual species shift would be likely to be similar with the situations from August to January in Fig 5.

**Zooplankton community diversity and biomass.** The zooplankton species were mainly composed by small individuals (i.e., protozoans and rotifers). Specifically, 35 protozoans and 43 rotifers were identified respectively, accounting for 32.4% and 39.8% of the total species. 18 cladocerans and 12 copepods species were identified, accounting for 16.7% and 11.1% of the total species (Fig 6). In general, the numbers of species showed an obviously seasonal characteristic, being higher in April and August than that in January generally. There were also of spatial characteristic with lower species number in the Yangtze River than that in the tributaries. ZX River showed the greatest variation of species numbers, i.e., ZX River has registered 24 ± 9 species in April, 27 ± 6 species in August and 6 ± 11 in January, respectively.

The variation of biodiversity index (*H'*) was consistent with species number variations, showing generally lower values in Yangtze River than its tributaries. The *H'* in the Yangtze River ranged from 0.69 to 1.70 with a mean value of 1.19 ± 0.24 throughout the whole investigation period, reaching the maximum of 1.33 ± 0.26 in August and the minimum of 1.12 ± 0.08 in April. The *H'* in tributaries recorded larger variations, ranging from 0 to 2.65 with a mean value of 1.67 ± 1.03 through the year. It recorded the largest values in April (2.13 ± 0.45) and the smallest in January (0.78 ± 1.36). The species richness index (*D*) was roughly consistent with the *H'* variation. The evenness index (*J*) recorded the maximum value (0.95 ± 0.02) in January and the minimum value (0.52 ± 0.56) in April in LB River.

## Zooplankton communities and the associated environmental factors

The relationships between zooplankton community and environmental factors in water body of Wanzhou City were explored by CCA ordination. As mentioned earlier, CCA can obtain a

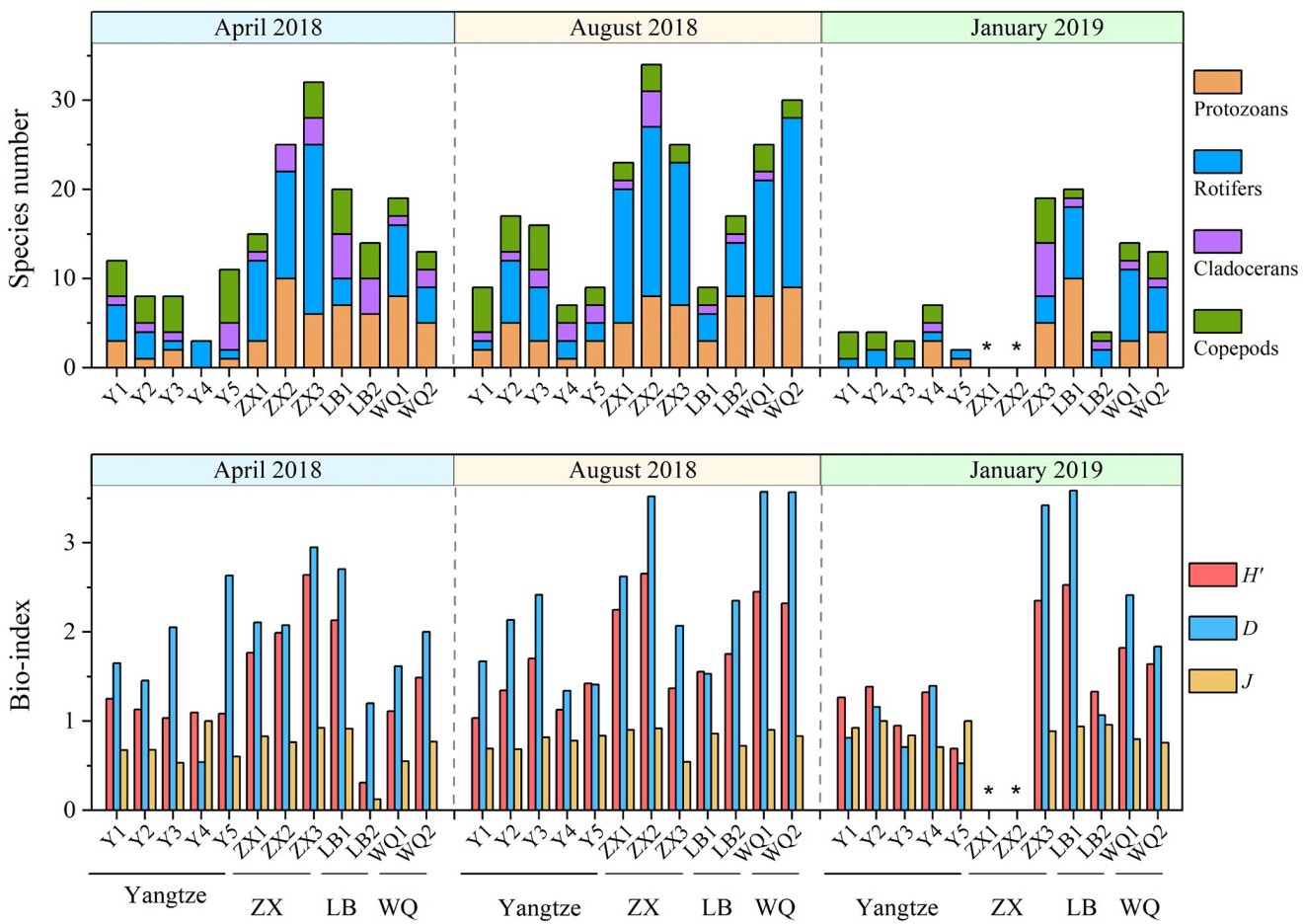

**Fig 6. Zooplankton species number and bio-index in the Yangtze River and its tributaries in Wanzhou.** Note: * not detected.

highly simplified affinity relationships among environmental parameters, zooplankton and ordination axes based on their distribution, contributing to elucidate relative contributions of significant environmental parameters to zooplankton assemblage variations (Fig 7). Based on the CCA performance (Table 2), the first four canonical axes were significantly correlated with zooplankton population changes ($p < 0.001$). The eigenvalues of the first two axes were 0.82 and 0.63, respectively, explaining 13.4% and 10.3% variations of response data (i.e., zooplankton species) and 34.4% and 26.4% variations of species-environment relations (i.e., fitted response data) in turn. The third and fourth axes explained 7.8% and 5.4% variations of response data, and 19.9% and 14.0% variations of species-environment relationships, respectively.

Six environmental parameters were detected to impose unique and significant effects on tempo-spatial variations of zooplankton assemblages (Fig 7 and Table 3). In details, $NO_{3^-}$-N accounted for the maximum variance (11.8%, $p = 0.001$), followed by T (9.5%, $p = 0.011$), $NO_{4^+}$-N (8.0%, $p = 0.001$), WL (5.8%, $p = 0.006$), $COD_{Mn}$ (3.8%, $p = 0.045$), respectively. Among the environmental parameters, $NO_{3^-}$-N and WL were positively correlated with the CCA axis 1 (CCA 1), whereas $NO_{4^+}$-N was negatively projected on the CCA 1. T and $COD_{Mn}$ were negatively correlated with the CCA axis 2 (CCA 2), whereas $NO_{4^+}$-N could be projected on positive end of CCA 2. All the aforementioned factors showed a strong affinity to

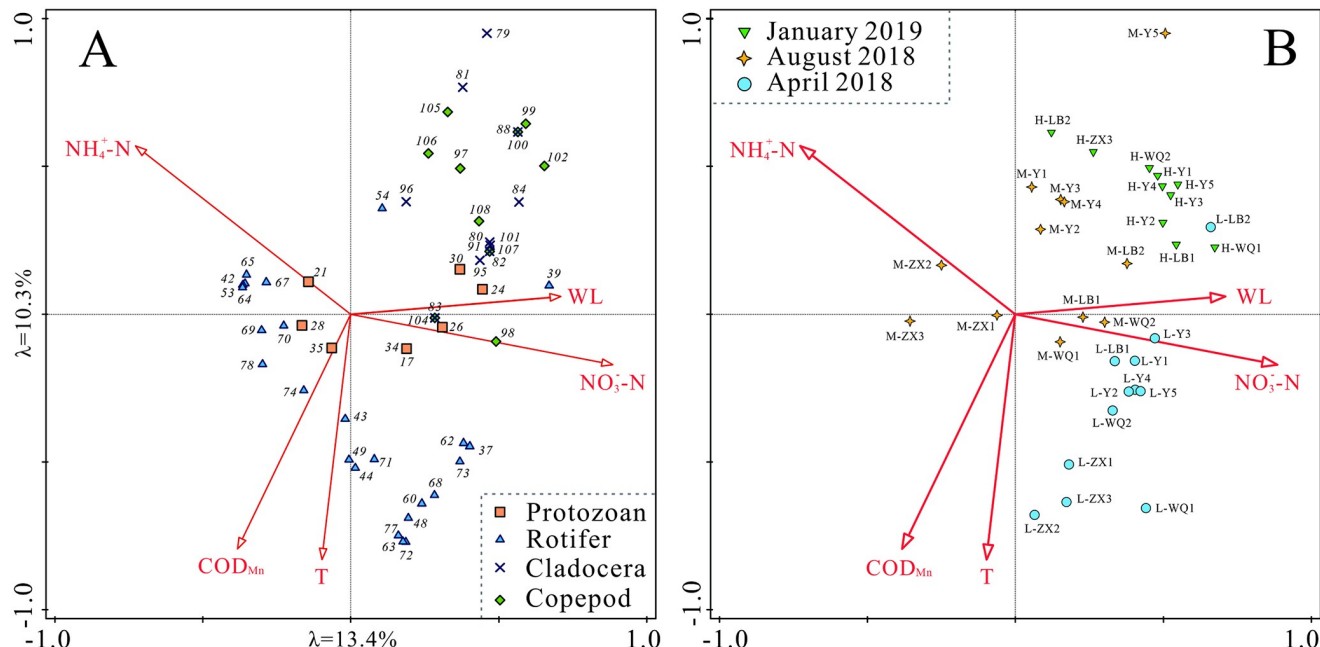

**Fig 7. CCA ordination plot based on zooplankton biomass data and environmental factors.** (A) zooplankton species vs. environmental factors, (B) samples vs. environmental factors. The number in (A) denoted for species, corresponding to that in S1 Table, the symbol in (B) expressed in form of "month-site".

first four axes of CCA with a contribution of 38.9% (Table 3), indicating that they were meaningful explanatory variables for the observed gradient along the CCA axes. Considering the projected positions from a specific specie symbol to an arrow of specific environmental factor, it can be used to observe an ordination trend along the environmental factor [24]. Nearly all of

**Table 2. Summary statistics for CCA performance between environmental factors and zooplankton.**

| Statistic | Axis 1 | Axis 2 | Axis 3 | Axis 4 |
|---|---|---|---|---|
| Eigenvalues | 0.82 | 0.63 | 0.47 | 0.33 |
| Pseudo-canonical correlations | 0.96 | 0.92 | 0.84 | 0.82 |
| Cumulative percentage variance | | | | |
| of response data | 13.4 | 23.7 | 31.5 | 36.9 |
| of fitted response data | 34.4 | 60.8 | 80.7 | 94.7 |
| Sum of all eigenvalues/Sum of all canonical eigenvalues | 6.09/2.37 | | | |
| Test of significance of first canonical axis: | $F = 4.33$, $p = 0.002$ | | | |
| Test of significance of all canonical axes | $F = 3.57$, $p = 0.001$ | | | |

**Table 3. The forward selection results of environmental factors in CCA.**

| Parameters | Explains (%) | Contribution (%) | Pseudo-F | $p$ |
|---|---|---|---|---|
| $NO_3^- - N$ | 11.8 | 21.8 | 4.3 | 0.001 |
| T | 9.5 | 17.6 | 3.8 | 0.011 |
| $NO_4^+ - N$ | 8.0 | 14.7 | 3.4 | 0.001 |
| WL | 5.8 | 10.7 | 2.6 | 0.006 |
| $COD_{Mn}$ | 3.8 | 7.0 | 1.7 | 0.045 |

cladocerans and copepods appeared at the positive end of $NO_{3-}$-N and WL, indicating those mesozooplanktons were favored of high $NO_{3-}$-N and WL (Fig 7A). Some rotifers, such as *Polyarthra dolichoptera*, *Brachionus angularis*, *Keratella valga*, *A. priodonta*, *P. vulgaris*, *Synchaeta pectinate*, *Synchaeta* sp., *Filinia longiseta*, *Trichocerca rousseleti* ect. projected the positive end of $NO_{4+}$-N, implying they were able to tolerate habitats with high $NO_{4+}$-N concentration, as well as some protozoans, e.g., *Chilodonella* sp., *Vorticella* sp., Ciliate. It can be also seen that cladocerans and copepods were more sensitive to ammonia comparing with some rotifers and protozoans. Similarly, a group of rotifers were clustered around the arrows of T and $COD_{Mn}$ (e.g., *Cephalodella gibba*, *E. senta*, *F. cornuta*, *Brachionus quadridentatus cluniorbicularis*, *Monostyla bulla*, *Synchaeta oblonga*, *B. diversicornis*, *B. quadridentatus melheni*, *Cephalodella exigua* ect.), suggesting those species displayed remarkable adaptive characteristics with warmer water and high $COD_{Mn}$ content comparing with cladocerans and copepods.

CCA also revealed the tempo-spatial features of samples at different seasons (Fig 7B). In details, water samples collected in April and January recorded higher $NO_{3-}$-N concentration, whereas water samples collected in August were characterized with higher $NO_{4+}$-N concentration generally. Along CCA 2, samples in August were projected to high optima values of T and $COD_{Mn}$ when water level was the lowest, especially the samples of ZX Rivers. Similar tempo-spatial features could also be identified according to Table 1. The information of Fig 7 could depend our understandings regarding biological characters of zooplankton effected by a specific environmental factor.

## Discussion

### The tempo-spatial differences of environmental factors and zooplankton

In general, the water quality was obviously better in the Yangtze River than its tributaries in Wanzhou section of the TGRA (Table 1 and Fig 2). That may be related to the water discharge of rivers flowing through the urban area. The Yangtze River, as the longest river in China, is of high flow capacity, leading to a certain dilution effect on nutrient pollutants in water body even during the lowest flow [25]. However, the tributaries have less water discharge, thus it's easier to enrich pollutants under the influences of anthropogenic activities (e.g., urban sewage, agricultural fertilizers in the upstream catchment) and resulted in eutrophication. The TSI of Yangtze River in August was significantly higher than the other two seasons (Fig 2), may be associated with the hydro-climatological processes. In details, the rainfall resulted high runoff usually brings more carbon-, nitrogen- and phosphorus-relating pollutants into the Yangtze River, ultimately increasing the TSI in August. It was supported by the positive relationship between TSI and TSS in this study (r = 0.38, $p$ = 0.022, not shown), implying the effects of high runoff on pollutant loads.

One of the most outstanding features regarding the tempo-spatial variations of zooplankton was that rotifers were relatively higher in April and August when temperature was also high, and copepods accounted for the highest proportion in January when temperature was the lowest. This may be associated with multiple factors, such as biological characteristics of zooplankton, predator-prey relations in food web, environment factors and so on. Specifically, cladocerans (e.g., *Daphnia*) and copepods which are of relatively large individual size were documented to favor cold stenothermal water body [26, 27]. The well visible cladocerans and copepods suffer more predation pressure from fish than smaller species [28, 29], juvenile zooplanktons and rotifers may thus be possible to increase under fish predation [30]. Meanwhile fish densities were much higher in summer at LWL than that in winter at HWL [31], as well as the fish activities, implying a high intensity of predation pressure on zooplankton. Those factors may theoretically affect the seasonal variations of zooplankton. However, we focus on the

relationships between environment factors and zooplankton assemblages in this study (more details were discussed below). Definitely, environment factors can play direct and indirect roles on zooplankton. For instance, $NO_{3^-}$-N, T, WL, $NO_{4^+}$-N $COD_{Mn}$ exhibited significant correlations with density or biomass of zooplankton categories (Fig 7), implying their potential influences on zooplankton.

## Relationship between zooplankton and dominant environmental factors

Again, the first axis was associated with $NO_{3^-}$-N, $NO_{4^+}$-N and WL, reflecting the hydrology and nutrient relating changes roughly (Fig 7), and the second axis was associated with T and $COD_{Mn}$ in water body, reflecting the seasonal characteristics. Plenty of research has suggested that those variables were the possible major environmental factors affecting structure of zooplankton fauna [10, 11, 32].

**Zooplankton community distribution on CCA 1.** $NO_{3^-}$-N, $NO_{4^+}$-N and WL have been widely reported to be important nutrient and hydrological variables that indirectly influence zooplankton dynamics by virtue of mediating phytoplankton growth indicated by Chl a in aquatic ecosystems [2, 33–35]. Those genera observed in this study, such as *Brachionus*, *Rotaria*, *Keratella*, *Paramecium*, *Schmackeria* and *Polyarthra*, are typical indicators of eutrophic water body where $NO_{3^-}$-N and $NO_{4^+}$-N concentrations are normally high [33, 36–39]. WL fluctuations in TGRA leads to the cyclical alterations of physicochemical properties of water environment and constantly changing habitats, especially nitrogen-relating nutrients recycling under WL fluctuating circumstances. Nitrogen-relating nutrients were not only of tempo-spatial differences (Table 1), but also bring about changes of zooplankton (Figs 7 and 8).

Nitrate and ammonia, as substantially existing forms of nitrogen in water body or WLFZ of TGRA, was served as one of the material foundations inducing algae bloom [7, 8], relating to exogenous input and endogenous release. The exogenous input mainly included atmospheric deposition, fertilizer use, soil organic carbon leaching, industrial pollution and sewage etc. [7, 40]. The endogenous release was involved to nitrogen transformation induced by biogeochemical processes, such as nitrification, denitrification, bio-assimilation, sedimentation and sediments release [7, 8, 41, 42]. Specifically, nitrogenous fertilizer, soil organic nitrogen and sewage were regarded as the dominant nitrogen sources of water body in TGRA [6, 7, 10, 43]

In spring at MWL, high $NO_{4^+}$-N concentration of water body may associate with intensively used nitrogenous fertilizers in agriculture, domestic sewage pollutions and a significant accumulation of $NO_{4^+}$-N generated by ammonification from organic nitrogen in WLFZ sediment under moderate intensity precipitation condition [7, 42]. Certain microzooplanktons became the dominant species, such as *A. priodonta*, *Synchaeta* sp., *P. vulgaris*, *T. rousseleti*, *B. angularis*, *K. valga*, *Chilodonella* sp., *Vorticella* sp. Those aforementioned species were more tolerant to high ammonia than most copepods and cladocerans, being similar with the observations of toxicology research [35]. In summer at LWL, more pollutants were washed into the water body due to high rainfall, leading to high TSS (Table 1) and the resulted higher amount of particulate nitrogen [44]. Meanwhile, higher temperature and moisture stimulated microbial activity and accelerated soil nitrogen transformation from organic matter or ammonia to nitrate by virtue of mineralization, ammonification and nitrification in summer [42], ultimately resulted in high $NO_{3^-}$-N concentrations in water body through nitrogen leaching. Thus, zooplankton has shifted to nitrate-tolerant species, such as *Encentrum* sp., *E. senta*, *F. cornuta*, *S. dorrii*, Cyclopoida Copepodid, *C. exigua*, *S. oblonga*, *P. erythrophthalma*, *B. quadridentatus*, *E. dilatata*, *S. forbesi*. In winter at HWL, submerged plants and soils in the submerged WLFZ and nitrification in the surface riverine water, together with domestic sewage,

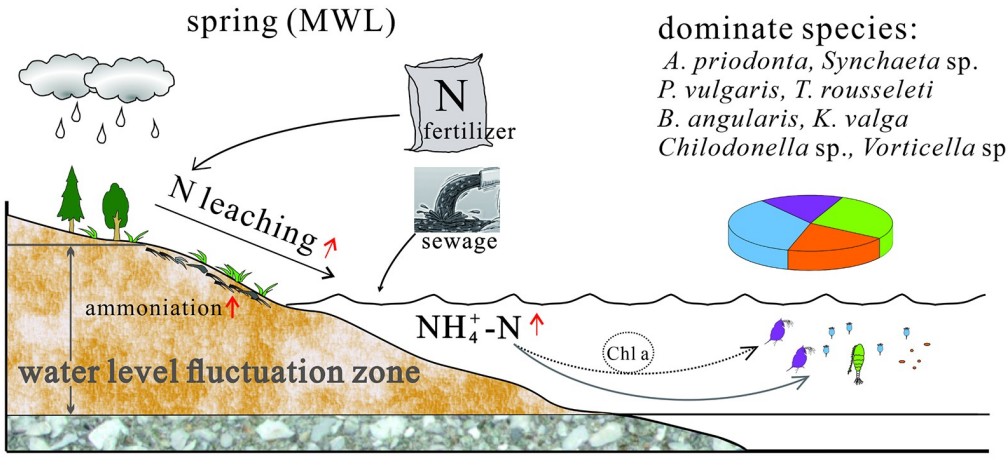

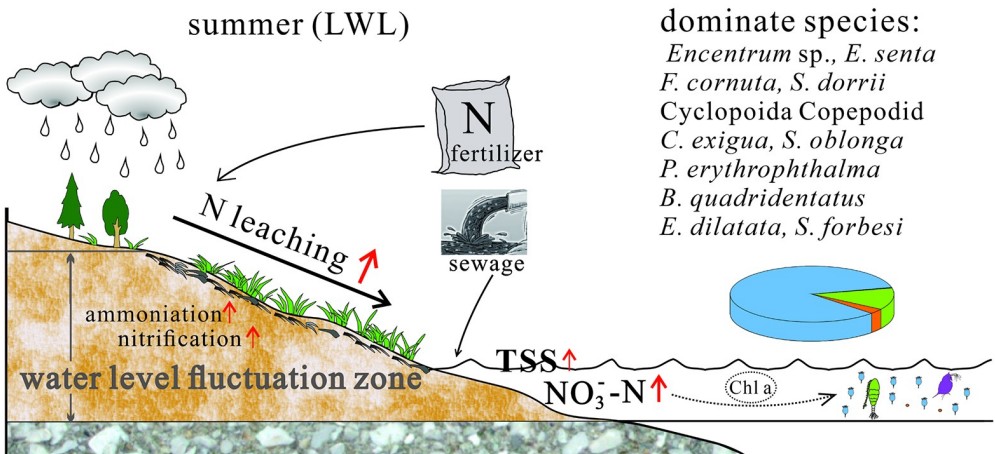

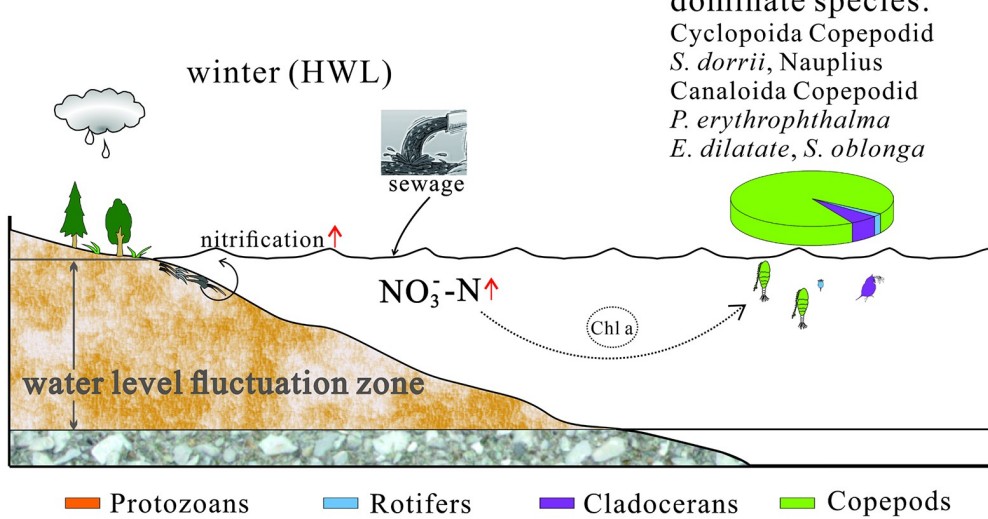

**Fig 8. The conceptual graph of relationships between hydrological variations and dominant zooplankton species.**
Only show the species with biomass proportion >5% and frequency ≥5 in each season.

led to a moderately high $NO_3^-$-N level in water body [7, 8, 45, 46]. Certain mesozooplanktons and larvae were much more popular under this environment, such as Cyclopoida Copepodid, *S. dorrii*, Nauplius, Canaloida Copepodid, *Bosmina coregoni*, as well as some nitrate tolerant microzooplanktons, e.g., *P. erythrophthalma*, *E. dilatate* and *S. oblonga*. In conclusion, aforementioned species shifts were possibly the comprehensive results of water environments, including the nutrient ($NO_3^-$-N, $NO_4^+$-N), seasonal characters (e.g., T and precipitation), water level fluctuations, and other unmentioned factors.

**Zooplankton community distribution on CCA 1.** Our research showed that T and $COD_{Mn}$ (mainly indicating organic pollution) was positively correlated with some microzooplankton and negatively correlated with mesozooplankton (i.e., cladocerans and copepods), similar with other studies [47]. Temperature may impose various effects on zooplankton, directly accelerate smaller sized species proliferation, causing planktons were more dominant by fast growing and small sizes species [3], such as protozoans and rotifers. Rotifers usually display r-strategy life histories and are adaptable to environmental disturbances [48]. Meanwhile, temperature can restructure food webs, changes food resources (e.g., phytoplankton) and predators (e.g., fish) quantitively and qualitatively, imposing indirect effects on zooplankton ultimately [49–51]. In this study, T and Chl a show highly significantly positive correlation (r = 0.46, *p* = 0.006, not shown in Fig 7 because of high collinearity with T and $COD_{Mn}$), implying indirect effects of T according the predator-prey relationship mediated by Chl a.

The microzooplankton (e.g., such as *C. gibba*, *E. senta*, *F. cornuta*, *B. quadridentatus cluniorbicularis* ect.) that adapted to high $COD_{Mn}$ condition in this study (Fig 7) have been widely reported to indicate high organic pollution [12, 38, 52] because the distribution or biomass of microzooplankton could be linked to organic pollutants significantly [47]. Normally, organic pollutants are available to be biodegraded by planktonic bacteria, resulting that the bacterial biomass increased with organic pollutants and further enhanced the growth of microzooplankton that prey on bacteria in water body [47, 53, 54]. Thus, organic pollutants indirectly modulated microzooplankton assemblages via predation relations. In addition, it has been widely accepted that the bacterial size organic pollutants can be directly consumed by microzooplankton [47]. In a word, increasing organic pollutants was beneficial to microzooplankton reproductions via microbial food web in aquatic ecosystems, thus it reasonably explained the positive correlation between microzooplankton and $COD_{Mn}$ in our study. Some researchers argued that rotifers showed strong resistance to nutrient-rich water environment, on the contrary, cladocerans prefer medium or poor nutrient water environment [19, 55]. Thus, in a word, microzooplankton may be more popular in water body with high $COD_{Mn}$ concentration comparing with mesozooplankton generally.

Further analysis was performed by using the index $P_{micro/meso}$ (i.e., the proportion between the microzooplankton and mesozooplankton) to reveal the relationship between individual size of zooplankton assemblage and T and/or $COD_{Mn}$. A good linear positive relationship between log transformed $P_{micro/meso}$ and T was detected, as well as log transformed $P_{micro/meso}$ and $COD_{Mn}$ (Fig 9). The linear relationship was even better based on density than that based on biomass. It clearly expressed a miniaturized trend of zooplankton individual size in virtue of species shift as T or $COD_{Mn}$ increased. Thus, it implied that zooplankton could produce fundamental structure differentiation that effected the food webs in aquatic ecosystem under the scenarios of global warming or increasing organic pollutions in future. Furthermore, the $P_{micro/meso}$ can be taken as useful bioindicators of size structure of zooplankton, closely associating with specific environmental gradient, e.g., the trends of $COD_{Mn}$ and T in this study.

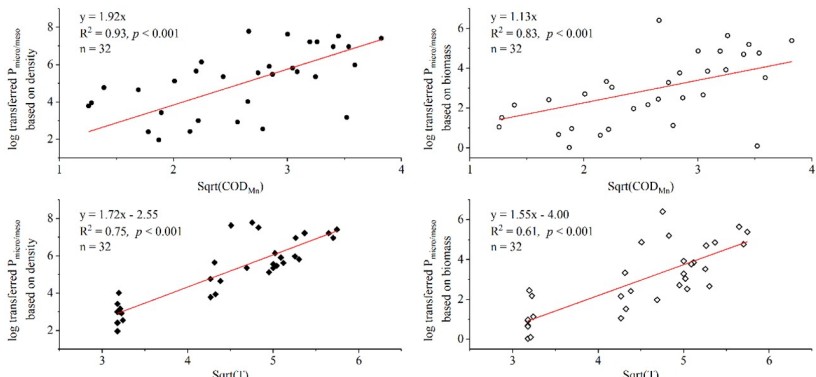

**Fig 9. The relationship between zooplankton size and key environmental parameters that were associated with CCA 2.** $P_{mico/meso}$ mirrored the portion of microzooplankton to mesozooplankton based on density and biomass data. $P_{mico/meso}$ were log transferred, and environmental parameters ($COD_{Mn}$ and T) were sqrt transferred. Four samples were excluded because the denominators of the ratios were 0.

## Ecological implication for climate changes and anthropogenic activities

In general, the vital environmental factors that modulated the zooplankton assemblages (including $NO_3{}^- $-N, T, $NO_4{}^+$-N, WL and $COD_{Mn}$) in this study were related with anthropogenic activities or climate change, possibly influencing aquatic organisms and being potential threats to aquatic ecosystem stability based on modern and paleo ecology research [56–59]. Ecosystem feedback to climate warming revealed a trend of enhancing carbon uptake in high precipitation regions based on meta-analysis [60], indicating that increasing organic carbon in water will be expected under the global warming by promoting the algae growth or under the effects of increasing sewage discharges. Those factors directly or indirectly reshaped the zooplankton community composition at expense of large-bodied species [32; this study], similar with the observation in marine aquatic ecosystem [61]. And nitrogen in water body of TGRA will be increased in anticipation of intensive human activities, for example, $NO_3{}^-$-N presented an increasing load of 6% every year during transport through the TGR, as well as other forms of nitrogen [8, 41]. All the factors will intertwine actively to produce significantly seasonal characteristics of water environment and zooplankton ultimately [7, 62, 63]. All in all, anthropogenic activities and climate change may lead to an increasing dominance of small-bodied zooplankton in the water body of TGRA, and thus reorganize biomass structure of food webs, imposing profound effects on the structure and functioning of aquatic communities and ecosystems [58, 64, 65]. In future, long-term successions of zooplankton within a long time-window of climate change and predator-prey relations that zooplankton involved in aquatic ecosystems need to draw more attentions. Those research results will help us to deepen our understandings on the comprehensive effects of climate change and anthropogenic activities imposing on aquatic ecosystems.

## Conclusions

An ecological investigation regarding zooplankton and the associated environmental factors was conducted in Wanzhou City of the TGRA during April 2018 to January 2019. Several conclusions can be made as following:

1. The TSI of the Yangtze River in Wanzhou City varied from 46.3±3.3 to 60.3±5.1, indicating that the water quality reached a state of mesotrophication to light eutrophication, while the

TSIs of its tributaries (i.e., ZX River, LB River and WQ River) varied from 60.2±5.6 to 71.4 ±5.1, implying moderate to hyper eutrophication of water quality status

2. A total of 108 species and subspecies of zooplankton in 69 genera were identified in this study, being of obvious tempo-spatial variations. In April, *A. priodonta*, Canaloida Copepodid and *S. dorrii* were dominant species in the Yangtze River, and *A. priodonta* was also the dominant species in the tributaries. In summer, *Encentrum* sp. accounted for the largest proportion in the Yangtze River and WQ River, whereas *F. cornuta* and *E. senta* were the most noticeable species in ZX River and LB River, respectively. In winter, Cyclopoida Copepodid, *S. dorrii* and *P. erythrophthalma* became the dominant species in the Yangtze River and its tributaries. Generally, rotifers and copepods prevailed in April, rotifers were absolutely dominant category in August, and copepods became the most popular category in January. The *H'* of zooplankton ranged from 0.69 to 1.70 with a mean value of 1.19 ± 0.24 in the Yangtze River, and it ranged from 0 to 2.65 with a mean value of 1.67 ± 1.03 in tributaries through the year.

3. $NO_{4+}$-N, T, $NO_{4+}$-N, WL and $COD_{Mn}$ imposed unique and significant effects on tempo-spatial variations of zooplankton assemblages based on CCA. $NO_{3-}$-N, $NO_{4+}$-N and WL were significantly associated with CCA 1 ($p < 0.01$), whereas T and $COD_{Mn}$ were significantly associated with CCA 2 ($p < 0.05$). Most of cladocerans and copepods appeared to favor high $NO_{3-}$-N and WL. Some rotifers or protozoans (e.g., *P. dolichoptera*, *B. angularis*, *K. valga*, *A. priodonta*, *P. vulgaris*, *S. pectinate*, *Synchaeta* sp., *F. longiseta*, *T. rousseleti*, *Chilodonella* sp., *Vorticella* sp., Ciliate ect.) were able to tolerate high $NO_{4+}$-N habitats, some rotifers (e.g., *C. gibba*, *E. senta*, *F. cornuta*, *B. quadridentatus cluniorbicularis*, *M. bulla*, *S. oblonga*, *B. diversicornis*, *B. quadridentatus melheni*, *C. exigua* ect.) displayed remarkable adaptive characteristics with higher T or $COD_{Mn}$ content comparing with cladocerans and copepods. The zooplankton species shifts may be partly associated with migration and transformation of nitrogen in the aquatic ecosystem under periodic WL fluctuation.

4. The $P_{micro/meso}$ can be taken as useful bioindicators of size structure of zooplankton, closely associating with environmental gradient of T and $COD_{Mn}$. The strongly positive relationship between $P_{micro/meso}$ and T ($p < 0.001$), and $P_{micro/meso}$ and $COD_{Mn}$ ($p < 0.001$), implied that zooplankton assemblage tended to miniaturize individual size via species shifts under high T and/or $COD_{Mn}$ conditions induced by human activities and global warming, being likely to reorganize biomass structure of the food webs and thus produce fundamental changes in aquatic ecosystem in future.

## Supporting information

**S1 Table. The species composition of zooplankton during the three seasons.** "+" denoted for the occurrence of species.
(DOCX)

## Acknowledgments

Tianfu Feng, as a driver, is gratefully acknowledged for his support in sampling.

## Author Contributions

**Conceptualization:** Bo Lan, Chi Zhu.

**Formal analysis:** Bo Lan, Yujing Huang.

**Funding acquisition:** Bo Lan.

**Investigation:** Bo Lan, Liping He.

**Methodology:** Bo Lan, Liping He, Yujing Huang, Chi Zhu.

**Resources:** Xianhua Guo.

**Validation:** Wenfeng Xu.

**Writing – original draft:** Bo Lan.

**Writing – review & editing:** Bo Lan, Liping He, Yujing Huang, Xianhua Guo, Wenfeng Xu, Chi Zhu.

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
