## [Decision Letter · Decision Letter 0]

22 Jul 2021

PONE-D-21-13276

Tempo-spatial variations of zooplankton communities in relation to environmental factors and the ecological implications: a case study in the hinterland of the Three Gorges Reservoir area, China

PLOS ONE

Dear Dr. ZHU,

Thank you for submitting your manuscript to PLOS ONE. After careful consideration, we feel that it has merit but does not fully meet PLOS ONE’s publication criteria as it currently stands. Therefore, we invite you to submit a revised version of the manuscript that addresses the points raised during the review process.

We look forward to receiving your revised manuscript.

Kind regards,

Hans-Uwe Dahms, Ph.D.

Academic Editor

PLOS ONE

Journal Requirements:

2. In your Methods section, please provide additional location information of the sampling sites, including geographic coordinates for the data set if available."

We note that one or more of the authors are employed by a commercial company: "Jiangsu Environmental Engineering Technology Co. LTD"

Additional Editor Comments (if provided):

This contribution analyses the water pollution status in urban aquatic ecosystem of Wanzhou City, spatio-temporal distribution

patterns of zooplankton and investigated the associated ecological successions and key environmental factors shaping the

community structures. The manuscript is acceptably written and analysed the data in appropriate ways. Following Reviewer 1,

I have the following comments to improve the quality of manuscript:

i) In abstract the authors can include quantified data when summarising salient observations.

ii) Detailed methodologies of some of the major parameters like Chl-a can be explained in materials and methods.

iii) For nutrient analysis, details on QA/QC procedures, calibration, detection limits can be given.

iv) The authors need to be explained briefly CCA results on the manuscript.

v) Conclusion part need to be restructured and main results.

vi) ‘n’ values need to be provided for Fig. 9.

Vii) The reported literature need to include the recent findings from 2020 and 2021.

Reviewers' comments:

Reviewer's Responses to Questions

**Comments to the Author**

1. Is the manuscript technically sound, and do the data support the conclusions?

Reviewer #1: Yes

2. Has the statistical analysis been performed appropriately and rigorously? 

Reviewer #1: Yes

3. Have the authors made all data underlying the findings in their manuscript fully available?

Reviewer #1: Yes

4. Is the manuscript presented in an intelligible fashion and written in standard English?

Reviewer #1: Yes

5. Review Comments to the Author

Reviewer #1: The manuscript reports the present situation of water pollution status in urban aquatic ecosystem of Wanzhou City, spatio-temporal distribution patterns of zooplankton and investigated the associated ecological succession processes and the key environmental factors shaping the community structures. The manuscript is well written and analysed the data in detailed way. I have the following comments to improve the quality of manuscript:

i) In abstract the authors can include some quantifiable data when they summarise their salient observations

ii) Detailed methodologies of some of the major parameters like Chl-a can be explained in materials and methods section

iii) For nutrient analysis, details on QA/QC procedures, calibration, detection limits can be given

iv) The authors can explained briefly CCA results on the manuscript

v) Conclusion part can be restructured and explain only the brief result; not introduction and methodology.

vi) ‘n’ values can be given in the Fig. 9

6. PLOS authors have the option to publish the peer review history of their article (what does this mean?). If published, this will include your full peer review and any attached files.

Reviewer #1: No

---

## [Author Response · Author response to Decision Letter 0]

31 Jul 2021

Journal Requirements:

Answer: Yes, we have followed the templates to revise the style of the manuscript and checked it again.

2. In your Methods section, please provide additional location information of the sampling sites, including geographic coordinates for the data set if available."

Answer: In fact, we have provided the geographic coordinates of all sampling sites in Fig 1. We add more detailed information in line 115-117.

Answer: Actually, we are pretty sure that there is no need to declare the permits of the field investigations in the kind of public area in China. The local governments will not prevent this investigations, in contrast, they quite welcome it because this kind of work can provide the status of the water quality and biocenosis, being helpful to the ecological managements. And our fundings from Chinese government are completely open, not classified type, thus the local governments tend to support this field investigation. It can be said that that this type of field research papers where the study areas is located in China don’t need to provide the permits for their work.

Answer: Done

We note that one or more of the authors are employed by a commercial company: "Jiangsu Environmental Engineering Technology Co. LTD"

Answer: Done as required, see the amended Funding Statement in the cover letter.

Answer: Done as required, see the amended Funding Statement in the cover letter.

Additional Editor Comments (if provided):

This contribution analyses the water pollution status in urban aquatic ecosystem of Wanzhou City, spatio-temporal distribution patterns of zooplankton and investigated the associated ecological successions and key environmental factors shaping the community structures. The manuscript is acceptably written and analysed the data in appropriate ways. Following Reviewer 1,

I have the following comments to improve the quality of manuscript:

i) In abstract the authors can include quantified data when summarising salient observations.

Answer: Done, see line 14-34.

ii) Detailed methodologies of some of the major parameters like Chl-a can be explained in materials and methods.

Answer: Done, see line 135-137.

iii) For nutrient analysis, details on QA/QC procedures, calibration, detection limits can be given.

Answer: Done, see line 137-143.

iv) The authors need to be explained briefly CCA results on the manuscript.

Answer: Done. We further modified part context of the CCA results as suggested, see line 268-305. 

v) Conclusion part need to be restructured and main results.

Answer: Done, see line 450-485.

vi) ‘n’ values need to be provided for Fig. 9.

Answer: Done.

Vii) The reported literature need to include the recent findings from 2020 and 2021.

Answer: Done.

---

## [Editor Report · Decision Letter 1]

4 Aug 2021

Tempo-spatial variations of zooplankton communities in relation to environmental factors and the ecological implications: a case study in the hinterland of the Three Gorges Reservoir area, China

PONE-D-21-13276R1

Dear Dr. Zhu,

We’re pleased to inform you that your manuscript has been judged scientifically suitable for publication and will be formally accepted for publication once it meets all outstanding technical requirements.

Kind regards,

Hans-Uwe Dahms, Ph.D.

Academic Editor

PLOS ONE
---

## [Editor Report · Acceptance letter]

9 Aug 2021

PONE-D-21-13276R1 

Tempo-spatial variations of zooplankton communities in relation to environmental factors and the ecological implications: a case study in the hinterland of the Three Gorges Reservoir area, China 

Dear Dr. Zhu:

I'm pleased to inform you that your manuscript has been deemed suitable for publication in PLOS ONE. Congratulations! Your manuscript is now with our production department. 

Kind regards, 

on behalf of

Dr. Hans-Uwe Dahms 

Academic Editor

PLOS ONE